# Locked-in syndrome in Sweden, an explorative study of persons who underwent rehabilitation: a cohort study

Kajsa Svernling, Marie Törnbom, Åsa Nordin, Katharina S Sunnerhagen

Institution of Neuroscience and Physiology, University of Gothenburg, Gothenburg, Sweden

**Correspondence to**
Professor Katharina S Sunnerhagen;
ks.sunnerhagen@neuro.gu.se

## ABSTRACT

**Objective**  Locked-in syndrome (LiS) is a rare condition, characterised by a complete paresis except for vertical eye movements and blinking with cognitive functions intact, commonly caused by ischaemia in the ventral pons. Previous studies have indicated that persons with LiS can live on for many years and have a good quality of life (QoL). To our knowledge, LiS has never been studied in Sweden. The aim was to explore LiS in Sweden; describing population characteristics, living situation, mortality/cause of death and health-related QoL/impact on participation.
**Design**  Explorative, nationwide study with two parts (quantitative and qualitative). Persons registered in the national quality register WebRehab during 2007–2014 were eligible.
**Participants**  Ten persons were identified in part 1, four participated in part 2. During part 1, data were collected from WebRehab, medical charts and registers, while questionnaires and interviews were used during part 2.
**Results**  Seven out of 10 were men, median age at onset was 49 years and the cause of LiS was in all cases stroke, 70% of which ischaemic. Three were deceased with a median time of survival of 1.9 years. Seven were still alive, with a median time elapsed since onset of 5.8 years. Three participants experienced good QoL. Information, respect from professionals and more specialised technical devices were three areas containing unfulfilled needs.
**Conclusion**  This was the first study conducted in Sweden and the characteristics of this population were like those studied abroad. In this study, the persons with LiS who were interviewed expressed the need for proper care, appropriate technical aids and a supportive environment in order to have QoL. However, there is still much room for improvements.

## Strengths and limitations of this study

► The focus of this study was to explore and describe locked-in syndrome (LiS), which has to our knowledge never been studied in Sweden. The study population constituted the whole population of patients with LiS receiving in-patient rehabilitation between 2007 and 2014.
► We managed to interview four participants of the study population with help from others, and this contributed substantially to the contents of the study.
► This study is limited by its small number of patients and the results can only be indicative and cannot be generalised. Worth noting is that most studies are fewer than 10 or even single cases and as such, this study does not stand out as small.
► This study was performed in the Swedish context and the results relative to other cultural contexts need consideration.

## INTRODUCTION

Locked-in Syndrome (LiS) is a state in which a patient is both quadriplegic and paralysed up to the lower cranial nerves but conscious and with retained control of vertical eye movement and eyelids. LiS is usually caused by a brain stem lesion, most commonly a ventral pontine lesion that interrupts the descending pyramidal tracts.[1–3] The lesion is often a result of an ischaemic stroke due to thrombosis in the basilar artery, but can also be caused by haemorrhages, trauma, tumours or ischaemia due to hypotension.[4–6] In rare cases, the cause is metabolic or infectious.[7]

The American Congress of Rehabilitation Medicine[8] recommends neurobehavioural criteria to be used when diagnosing LiS. The criteria are (1) eye opening well sustained, (2) basic cognitive abilities preserved (clinical examination), (3) severe hypophonia or aphonia on clinical examination, (4) quadriplegia/quadriparesis on clinical examination and (5) communication primarily through eye movements or through blinking. The LiS can vary in clinical presentation. Classic LiS is defined as a fully paralysed patient with intact vertical eye movements and movement in the eyelid. IncompleteLiS is similar to classic LiS but with remnants of motor functions beyond those of the classic variant, usually some movement in fingers and/ or toes. Total LiS is defined as total immobility, the use of electroencephalography is then necessary to ascertain consciousness. Regarding duration, LiS can be chronic or transient. In the latter, the patients recover so that they not anymore fulfil the criteria

for LIS, but still might have (often substantial) sequelae due to their stroke.

The view on prognosis of LiS has shifted a lot through the years. When LiS first became a subject of study, the opinion was that acute mortality was high,[2] with nearly no long-term survivors.[9] Since then opinion has shifted, the numbers on mortality still vary between studies but the overall view on survival is more positive. If the patient medically stabilises and survives the first year, 5-year survival may be up to 81%–86%,[5 10] and some patients survive for decades.[7]

Patients with chronic LiS often continue to have highly impaired motor function, even if some improvement is possible.[3 10 11] Among other things, the impairments lead to them becoming dependent in the activities of daily living (ADL) (self-care, etc). Tetraplegia, along with impairments in breathing patterns, also means that respiratory complications are common.[3] Most patients living with LiS learn to communicate in some way.[7 10 12]

Studies on quality of life (QoL) have shown that measured with scales that include motor impairment, LiS patients show lower QoL than healthy controls.[13] But measured using scales that do not include motor impairment, their QoL is not significantly altered.[13] Mild and moderate depression is more common in LiS patients than in healthy controls.[13] It is common for patients to be more emotionally sensitive and experience involuntary crying or laughter after the onset of LiS, compared with before,[7] a known problem after injuries to the brainstem.[14]

Early medical stabilisation and early rehabilitation improves prognosis[4 10] and a correct and early diagnosis is essential to minimise suffering and enable proper care.[6] When caused by an ischaemic stroke, early stroke treatment such as anticoagulation and treatment with tissue plasminogen activator could enhance the possibilities of a greater recovery.[15 16]

Participation is an aspect of disability which is dependent on both personal and environmental factors, as shown in the International Classification of Functioning, Disability and Health,[17] which is why it becomes an important issue to discuss. Neither autonomy nor participation is static; they are values that can differ throughout life,[18] which means it can be reduced in one area but still be high in others. This becomes relevant in the topic of LiS, when motor function is low but cognitive function is high.

Severely disabled persons report good QoL despite their serious conditions, which is referred to as the disability paradox.[19] Despite researchers' belief that the disability paradox was real, it was only when Ubel *et al* investigated potential sources of error and explanations of this good QoL that they were able to conclude that it did indeed exist.[20] This emphasises the importance of caution for physicians and significant others when forced to make important decisions for their patients or next of kin, and the importance of not easily assuming a low life satisfaction.

The incidence of LiS in Sweden is unknown. Prior to 2015, there was no international classifications of disease (ICD)-code for LiS in ICD-10-SE (version 10, Swedish), which means the National Board of Health and Welfare has no statistics on the syndrome. LiS is not reported in the national quality register for stroke care, Swedish Stroke Registry(Riks-stroke).[21] One registry in Sweden, WebRehab, offers the possibility to report level of consciousness and also report LiS.[22] All the rehabilitation medicine clinics in Sweden participate in this quality register. There is a European Network for LiS patients but Sweden is not represented.[23]

To our knowledge, LiS has never been researched in Sweden. Previous studies have suggested that rehabilitation in these patients could improve if care was centralised and given by a skilled, professional team[3] and that early, intensive rehabilitation improves prognosis.[10] Therefore, there is a need for further research exploring LiS in Sweden, assessing possibilities and needs of that patient population. The aim of this study was to explore and describe LiS in Sweden with the purpose of gaining a better understanding of the life situation for this group of patients. Internationally, the information on LiS is also sparse, and in particular the long-term situation for them. This study will contribute some more information regarding LiS.

## MATERIAL AND METHODS

This study was performed with material from the registers WebRehab, medical charts, the Swedish Tax Agency's population register and The National Board of Health and Welfare's registry on Cause of Death (part 1) and from questionnaires and interviews (part 2).

WebRehab is a National Quality Registry in Rehabilitation Medicine. Twenty-three rehabilitation units in Sweden contribute to the database, representing all 21 counties of Sweden. The rehabilitation units report data from the rehabilitation period, including admission and discharge, and from a 1-year follow-up.[22] WebRehab is certified at level 2 which is the second highest certification level.

### Study population

Eligible for this study were persons registered in WebRehab between 2007 and 2014 for whom the level of consciousness[24] was reported. All were diagnosed with LiS (assessed by a specialist in rehabilitation medicine in collaboration with neurologist) and had been admitted for in-patient rehabilitation in specialised rehabilitation medicine clinics. In WebRehab, the following is stated as for LiS: the patient has tetraplegia and cannot use speech to communicate but is fully preserved cognitively, there is retained eye opening please exclude bilateral ptos that may complicate the condition), aphonia or hypophonia, tetraplegia or tetraparesis. There is also a question regarding communication (adequate yes/no

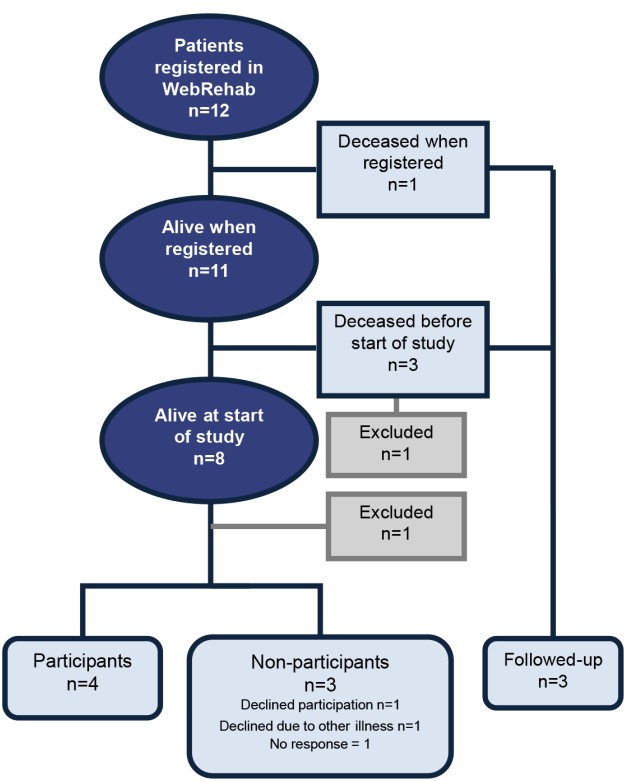

**Figure 1** Flow chart of the study. One person was excluded after the validation process due to not meeting the inclusion criteria (locked-in syndrome) and one person due to not having a valid personal identity number.

using vertical or horizontal eye movements alternatively eye blinking.

Twelve persons were identified from eight different hospitals. One person was excluded after the validation process due to not meeting the inclusion criteria and one person due to not having a valid personal identity number. For details, see figure 1.

## Procedure

It is stated that results will be used for a scientific publication as well as possible presentations at conferences. This is also clearly stated (as required by law) in the information letter to the presumptive study participants. They (or their next of kin) have given written informed consent, as well as oral, to publication.

In part 1, data from medical charts were gathered for validation and quality control of a quality register. No informed consent from the study participants is required as per Swedish law. In part 2, informed, written consent was obtained from all participants or their representative. This included consent for publication.

During part 1, data were collected from registers. The identity of the included study participants was obtained from WebRehab's database. The Functional Independence Measure (FIM) measures level of disability and independence in two scales, one motor scale and one cognitive scale[25] and was analysed for description of ADL functions and dependency.

To make certain that the data in WebRehab had been entered correctly (a validation), the medical charts were checked. To obtain charts, care units were contacted, first by letter, then reminders were sent by email, and finally, attempts were made to reach persons in charge by phone.

Data from the Swedish Tax Agency's population register were collected to investigate how many of the study persons were still alive and to obtain their addresses and contact information.

Data on cause of death and date of death were obtained from The National Board of Health and Welfare's registry on Cause of Death. Both the application and the communication following the application were written and handled by the first author.

During part 2, questionnaires and interviews were performed. An information letter along with the questionnaires was sent to all study persons still alive. If no response, a reminder was sent by letter and thereafter the attempts to reach the study persons or proxy was by phone. If the study persons wanted to participate, they were asked to send in the questionnaires or contact the author.

Four study persons who agreed to participate were visited for a personal, structured interview. The interviews were conducted in the participants' homes and were recorded and transcribed. The first author performed all interviews. During the interviews, a proxy gave some answers or a personal assistant and the rest were given by the participant and translated by next of kin. After the interview, the participant confirmed that the information given was correct, both the information given in the interview and in the questionnaires. One study participant was not able to perform a personal interview but participated through a telephone interview with her representative.

## Questionnaires

The different questionnaires used were Stroke Impact Scale (SIS-V.3.0),[26] RAND-36, Impact on Participation and Autonomy (IPA-E)[27] and EuroQol-5 dimensions (EQ-5D) briefly described below.

### Stroke Impact Scale-V.3.0

To assess health status a Swedish version of the SIS-V3.0[28] was used. The SIS is developed for, validated, and shown to be reliable for patients with stroke.[26 29]

### RAND-36

RAND-36 is a survey instrument that assesses health-related Quality of Life (HRQoL). The Swedish version of RAND-36 is a modern translation of the Short form Health Survey-36, but it is similar enough to allow comparisons. It is reliable and valid for measures on HRQoL in patients with stroke.[30]

### Impact on Participation and Autonomy

IPA-E was used to assess and measure IPA-E. The original version of IPA has good validity and reliability[31] IPA-E includes five domains: autonomy indoors, family role,

**Table 1** Characteristics of the study population

| | n=10 |
|---|---|
| Study persons | |
| Alive | 7 |
| Deceased | 3 |
| Cause of LiS | |
| Ischaemic stroke | 7 |
| Haemorrhagic stroke | 3 |
| Classification of LiS at admittance for rehabilitation | |
| Classic | 6 |
| Incomplete | 4 |
| Previous CVD/VRF | 6 |
| Total in-hospital stay median (range) | 216 days (150–627) |
| Length of stay in rehab median (range) | 151 days (63–345) |
| Age at onset median (range) | 49 years (22–67) |
| Survival* median (range) | 1.9 years (1.5–2.3) |
| Time since onset† median (range) | 5.9 years (2.3–8.1) |

*Deceased study persons only included.
†Alive study persons only included.
CVD, cardiovascular disease; LiS, locked-in syndrome; VRF, vascular risk factors.

autonomy outdoors, social life and relationships, and work and education.[27]

### EuroQol-5 dimensions

To assess HRQoL, a Swedish version of the questionnaire EQ-5D was used. EQ-5D is a standardised, validated health questionnaire developed by the EuroQol Group Association.[32] It is intended to be used for self-completion in postal surveys, interviews and clinical practice.[33] The EQ-5D is a valid and reliable measure of HRQoL after stroke.[33] EQ-5D assesses health in five dimensions: mobility, self-care, usual activities, pain/discomfort and anxiety/depression.

### Statistical methods

For statistical analyses, IBM SPSS Statistics V.21 was used. Mainly descriptive statistics with mean and median values were used for the demographic data.

### RESULTS
### Population characteristics

The characteristics of the study population are summarised in table 1.

Out of 10 study persons, there were three women and seven men; age at onset varied from 22 years to 67 years with a median of 49 years. In six of the cases of ischaemic stroke, the underlying cause was a basilar thrombosis, and the seventh case a vertebral artery dissection. Six persons had cardiovascular disease or documented vascular risk factors. The most common vascular risk factor was

hypertension. The patients were admitted to intensive care and then at stroke units before the rehabilitation period. The length of stay in hospital before rehab varied between the hospitals and was not always dependent on the medical needs of the patient. Nine of the study persons experienced respiratory complications during hospitalisation.

No change in ADL dependency in motor scale domains was seen in any of the study persons when measured with FIM at admission to rehab and discharge; all were completely dependent in all domains. However, the social-cognitive part of FIM improved with rehabilitation since communication was improved (which has an impact in these items).

Seven persons from the total population (n=10) were still alive at start of study and three persons were deceased. One person died during rehabilitation and the remaining two after the initial rehabilitation period, time from onset to date of death was 1.5, 1.9 and 2.3 years. The cause of death was pulmonary embolism, acute myocardial infarction and acute vascular disorders of the intestine. In none of the cases was the cause of death reported as being the result of respiratory complication due to LiS.

For details of all study persons and of participants, see tables 2 and 3.

### HRQoL and impact on participation

Results from the questionnaires are presented separately for each questionnaire. No mean scores were calculated due to the low number of participants in the follow-up. To put the participants' scores in perspective, values for reference populations are included in the tables.

The main finding from the questionnaires is that, although values vary between the follow-up participants, higher scores were seen in cognitive and mental domains (eg, SIS-V.3.0 Memory and Emotion and RAND-36 Mental Health) than in physical domains (eg, SIS-V.3.0 Strength, RAND-36 Physical Functioning and EQ-5D Mobility).

Individual scores on the SIS-V.3.0 are presented in table 4, on RAND-36 in table 5, on IPA-E in table 6 and on EQ-5D in table 7.

### Interviews
#### Information

In all but one interview, the follow-up participants described a lack of information about LiS and it was difficult to get information about technical devices or the follow-up procedure. They mentioned that they were sometimes forced to find technical aids and organise follow-up on their own. One participant, and a proxy, intended to write a book for persons with LiS about how to get information and support. This participant also found this information and support by becoming a member of a patient association.

When the follow-up participants were in intensive care, they felt they got better information than after discharge, although medical terms were sometimes difficult to understand.

**Table 2** All study persons

| | Study persons | | | | | | | | | |
|---|---|---|---|---|---|---|---|---|---|---|
| | 2 | 3 | 4 | 5 | 6 | 7 | 8 | 9 | 10 | 11 |
| Aetiology | Vascular (I) | Vascular (H) | Vascular (I) | Vascular (I) | Vascular (I) | Vascular (H) | Vascular (I) | Vascular (H) | Vascular (I) | Vascular (I) |
| Classification | Classic | Classic | Incomplete | Classic | Classic | Incomplete | Incomplete | Incomplete | Classic | Classic |
| Discharged to | Nursing home | Short-term care | Short-term care | Nursing home | Independent living with personal assistance | Independent living with personal assistance | Died during rehab | Independent living with personal assistance | Short-term care | Nursing home |
| Survival | 1 | 1.9 years | 1 | 2.3 years | 1 | 1 | 1.6 years | 1 | 1 | 1 |
| Cause of death (ICD-10-SE) | | Pulmonary embolism (I26.9) | | Acute vascular disorders of the intestine (K55.0) | | | Acute myocardial infarction (I21.9) | | | |
| Present form of residency | Nursing home | | Apartment with community-based support | | Independent living with personal assistance | Independent living with personal assistance | | Independent living with personal assistance | Independent living with personal assistance | Nursing home |

1, Still alive.
H, Haemorrhagic; I, Ischaemic; ICD-10-SE, International Classification of Disesae, version 10 (Swedish).

## Technical devices, technical adjustments and shortcomings

All used one or more wheelchair/s; alphabet boards and used or wanted to use an eye-tracking computer device. This kind of computer is expensive and was not available for everyone.

An alarm button which can be placed by the temple, adjusted hospital beds and lifts, tilting table, ceiling hoist, a walking frame with extra support and a sit-to-stand and transfer device were also mentioned as technical devices that the follow-up participants had access to. Two used bicycles for passive cycling. One person wanted a Functional Electric Stimulation-assisted training device, but the municipality refused funding. This person was also part of a customer test group for a device which combines eye-tracking technology with an electric wheelchair to enable control of the wheelchair with eye movements. He was promised money to purchase a wheelchair because of his help in developing the new technology.

Adjustments of homes were made, such as installing ramps, widening doors, thresholds and elevated toilet seats.

## Progress after rehabilitation and care

Several years after stroke, the follow-up participants' motor functions were still improving. They could communicate with the help of an alphabet board and used an eye-tracking computer device, or communicated mainly by blinking. One person had some oral communication including most vowels, some consonants and a few short words at discharge. At the time of assessment, he could control his jaw and tongue muscles and he had some movement in his fingers. Another could walk a few steps with support, lift his arms and move his head; he also had some function in his left hand, but was still completely dependent in ADL. Two persons could eat solid food and had no additional nutritional support. One person had an alarm button with a prerecorded sentence, which he could press by turning his head to attract attention. During the last year, he had learnt to shake his head and is currently practising nodding. All needed a lot of daily support and had personal assistants around the clock and sometimes two assistants at the same time.

## Likes and dislikes about care and authorities

The interviewed persons with LIS appreciated having phenomenal physiotherapists during the rehabilitation period. Every second week both an occupational therapist and a physiotherapist visited one participant, she still wanted a lot more additional training on top of these visits. Some said contacts with authorities mostly worked well, although it could be time-consuming, but one argued that daily contacts with caregivers worked less well. One participant felt unhappy because she was in a nursing home and could not live on her own terms but was granted a flat of her own in due time.

Some follow-up participants/proxies were offered support by a counsellor and were pleased with this. One partner talked about the difficult mourning process after

**Table 3** Assistance to and communication type of participants

| | Followed up patients | | | |
| | 2 | 4 | 6 | 10 |
|---|---|---|---|---|
| Time since locked-in syndrome | 8 years | 7.5 years | 5.5 years | 7 years |
| Living arrangement | Lives in a nursing home/home with society-based support. Joint common areas. | Lives in an apartment with society-based support. Lives alone. | Lives in a house. Lives with family. | Lives in a house. Lives with family. |
| Personal assistance | No personal assistants. Gets assistance from staff. | External personal assistants+extra assistance from staff when needed night-time. | External personal assistance+partner as paid assistant. | External personal assistance+partner as paid assistant. |
| Amount | Assistance from staff 168 hours/week | Assistance—168 hours/week Dual staffing—10.5 hours/week Total—178.5 hours/week. | Single staffing— 168 hours/week. | Assistance—168 hours/week Dual staffing—80.5 hours/week Total—248.5 hours/week. |
| Communication | Alphabet board, blinking. | Mainly alphabet board (blinking sometimes). | Oral, alphabet board (pointing). | Blinking, eye-tracking device. |
| Nutrition | Percutaneous endoscopic gastrostomy (PEG). | Oral, pureed/semisolid. | Oral, solid diet. | PEG, occasional treat by mouth. |
| Locomotion | Electric wheelchair. | Electric wheelchair. | Electric/manual wheelchair. Can walk a few steps with support. | Electric wheelchair. |

LiS. Nothing turned out the way she had wanted because of her husband's LiS. She was helped to think and reflect in a new way, such as what could be planned and done despite LiS. One partner reflected that during intense rehabilitation periods, one gets a lot of support from all specialists but then it ends, for example, appointments with a counsellor. One person wanted to meet a specialist doctor but felt he was not heard, having so many questions not answered by the local doctor.

### Attitudes from professionals
All interviewed experienced not being treated as adults, people talked to them slowly and loudly as if they were not able to understand anything which was frustrating.

**Table 4** Stroke impact scale results

| | Individual score | | | | Reference* |
| Domain | 2 | 4 | 6 | 10 | Mean (SD) |
|---|---|---|---|---|---|
| Strength | 25 | 37.5 | 25 | 0 | 71 (26.9) |
| Hand function | 0 | 0 | 30 | 0 | 81.1 (22.2) |
| Mobility | 0 | 0 | 19.4 | 5.6 | 77.6 (17.4) |
| ADL | 0 | 12.5 | 15 | 20 | 85 (21) |
| Emotion | 44.4 | 94.4 | 88.9 | 86.1 | 75 (28.5) |
| Memory | 100 | 42.9 | 100 | 71.4 | 75.6 (29.2) |
| Communication | 75 | 85.7 | 53.6 | 14.3 | 68.9 (34.5) |
| Social participation | 12.5 | 31.25 | 78.1 | 25 | 70.3 (27.5) |
| Stroke recovery | 20 | 50 | 100 | 10 | 68.4 (25.5) |

*Swedish stroke population, assessed 12 months after stroke.[28]
ADL, activities of daily living.

This could be aggravating if a person treating you did not know you well enough and/or did not have enough knowledge about the syndrome. A proxy said that just being able to nod your head led to a participant being treated with more respect.

### Family and friends
One participant had separated from her husband and had a complicated life situation. The others had mostly good family relationships, in two cases the wives worked as personal assistants. It was articulated that a relationship with a daughter had worsened after LiS and that this was the worst thing that had happened. Friends might have disappeared after LiS but the remaining ones were appreciated.

### Quality of life
All but one evaluated their current life situation. They mentioned feeling joy at being alive and having the ability to encourage others. Improving QoL could mean being granted more hours with personal assistants or getting in contact with a person with the same syndrome.

## DISCUSSION
### Findings
We identified 10 persons who had been diagnosed with LiS between 2007 and 2014 and investigated factors at onset, during the rehabilitation period, and after discharge. Seven of these persons were still alive at the start of the study and four of these participated. The common cause was stroke and most of these were ischaemic. According to medical charts, some of the study persons improved in their motor function and all are still improving, although

**Table 5 RAND-36 results**

| Domain | Individual score | | | | Reference (LiS)* Mean (SD) | Reference (stroke)† Mean (95% CI) |
|---|---|---|---|---|---|---|
| | 2 | 4 | 6 | 10 | | |
| Physical functioning | 0 | 0 | 5 | 0 | 0 (0) | 51.2 (44.3 to 58.1) |
| Role limitations due to physical health | 0 | 0 | 100 | 0 | 59.4 (32.6) | 14.7 (3.3 to 26.1) |
| Role limitations due to emotional problems | 33.3 | 100 | 100 | 100 | 75.0 (34.5) | 18.0 (10.2 to 25.8) |
| Vitality/energy-fatigue | 45 | 55 | 75 | 70 | 64.4 (24.6) | 42.9 (37.8 to 48.0) |
| Mental health/emotional well-being | 60 | 76 | 92 | 76 | 68 (19.6) | 62.7 (58.2 to 67.2) |
| Social functioning | 12.5 | 75 | 50 | 87.5 | 56.3 (34.1) | 55.2 (49.2 to 61.4) |
| Bodily pain | 80 | 90 | 100 | 67.5 | 82 (26.8) | 65.0 (57.9 to 72.1) |
| General health | 40 | 30 | 100 | 60 | 63.5 (33.0) | 58.2 (52.5 to 63.9) |

*Belgian LiS population, assessed more than 12 months after onset.[11]
†Swedish stroke population, assessed 2 years after day hospital rehabilitation for stroke.[37]
LiS, locked-in syndrome.

not in independency measured with the FIM motor scale. No major cognitive deficits were reported. Mortality in this population was 30% and mean survival time for the deceased was 1.9 years. For the study persons still alive, mean time since onset of LiS was 5.9 years. From the interviews, we identified areas with unfulfilled needs: information, respect and specialised technical aids. The data from the questionnaires were mostly in line with information from the interviews: those who expressed high QoL and high IPA-E in the interview, also scored high in domains not affected by motor impairments in the questionnaires.

### Strengths and limitations

Since this was a nationwide study and no selection was done, the number of patients might be considered few compared with previous studies abroad with study samples of around 20–30 persons.[5 13 34] One hypothesis could be that incidence numbers on LiS differ for some reason in Sweden. Another could be that incidence numbers are similar but not all persons with LiS in Sweden have been

**Table 6 Impact on participation and autonomy questionnaire results**

| Domain | Individual score | | | | Reference* Mean (SD) |
|---|---|---|---|---|---|
| | 2 | 4 | 6 | 10 | |
| Autonomy indoors | 4 | 3 | 0 | 2 | 0.96 (0.6) |
| Family role | 4 | 4 | 4 | 4 | 1.96 (1.1) |
| Autonomy outdoors | 4 | 3 | 2 | 3 | 2.35 (0.9) |
| Social life and relations | 3 | 2 | 3 | 3 | 1.48 (0.7) |
| Work and education | † | † | 1 | † | – |

*Iranian stroke population, assessed 5–36 months after their stroke.[38]
†Cannot be assessed, participant is not currently employed.

identified. This study population was identified through a register for patients having been admitted to rehabilitation medicine units, and therefore, did not include patients who died in intensive care units. Up until the end of 2014, WebRehab was, to our knowledge, the only register in Sweden that had statistics on LiS but as of 1 January 2015, there is an ICD-code in ICD-10-SE. What this will mean for the care of LiS-patients is hard to predict but reporting LiS will be possible for every care unit.

The questionnaires used are similar to each other in many ways, but all of them include unique aspects when compared with each other. EQ-5D is a quite rough instrument and physical functions have a large impact on the result. The value of using it in this particular study, where physical functions are very low, might therefore be worth discussing. In the EQ-5D, an index score under 0 is described as indicating a condition worse than death which is a problematic statement. For instance, three of our follow-up participants had an index score under 0, but none of them described their situation as worse than death.

When presenting the results of the questionnaires, values from reference populations are included in the tables, and our reasoning was to put our participants' scores in perspective. However, due to the low numbers of follow-up participants, no further comparisons can be made. The reference groups are all stroke populations, because all our participants had suffered a stroke and are included in this patient group.

The method chosen for this study was a quantitative, descriptive method containing analysis of register data, medical charts and structured interviews based on questionnaires. The reason we chose this approach instead of a qualitative method with unstructured interviews was the follow-up participants' limited ability to communicate effectively. This we believe would mean unstructured interviews would not give more information than structured interviews.

**Table 7** EuroQol-5 dimensions results

| Domain | Individual score | | | | Reference* Mean (95% CI) |
|---|---|---|---|---|---|
| | 2 | 4 | 6 | 10 | |
| Mobility | 3 | 3 | 3 | 3 | – |
| Self-care | 3 | 3 | 3 | 3 | – |
| Usual activities | 3 | 3 | 1 | 3 | – |
| Pain/discomfort | 3 | 2 | 1 | 2 | – |
| Anxiety/depression | 2 | 2 | 1 | 2 | – |
| Index value | −0.429 | −0.166 | 0.122 | −0.166 | 0.44 (0.38 to 0.50) |
| VAS | 0.02 | 0.5 | 1 | 0.4 | 62.7(58.8 to 66.6) |

*Swedish stroke population, assessed 2 years after day hospital rehabilitation for stroke.[37]
VAS, visual analogue scale.

By combining questionnaires with personal interviews, the follow-up participants had a chance to elaborate on their answers and opinions while still maintaining a standardised form of assessment with the questionnaires.

Since a proxy or a personal assistant was present and involved in the interviews, there is a risk of misinterpretations in translations or that things were not relayed in the way the participant intended. All follow-up participants were, therefore, asked if the information given was correct at the end of the interviews.

### Comparison with existing literature and guidelines

It seems consistent with previous knowledge that mortality is high during the first years after onset but when stabilised, persons with LiS may live for decades.[7] Since onset of LiS for our study population only goes back to 2007, real long-term survival cannot be commented on here.

The cause of death of the study persons was never a result of respiratory complications or problems with breathing, despite the high prevalence of respiratory complications during hospitalisation. This might be explained by proper care, which has prevented or successfully treated any respiratory problems, but it might also be explained by chance.

The importance of maintaining vital functions and that fatality in LiS has declined with improvements in quality of medicine has been discussed in previous studies.[6] In line with previous findings,[7 10 11] most of the study persons were middle aged with a median age of 49 years at onset. In line with our study, patients with chronic LiS have slim chances of major improvements in motor function.[10]

According to FIM, five persons improved in independency in cognitive domains. Expression and problem solving are the two areas with the least improvements, which might be explained by the poor communication skills of this patient group.

In many aspects, the results on the questionnaires varied between the participants but most of them scored high on domains measuring emotional and cognitive functions and low on physical domains and functions. This is in line with previous knowledge that QoL in persons with LiS can be high scoring in mental domains, while low scoring in physical domains.[13]

### Clinical and policy implications

Problems like nursing staff using complicated medical terminology or not getting enough information about technical and rehabilitation aids were brought up. As this patient group has limited communication skills, it seems important to fulfil all their needs for technical equipment to improve their QoL. All participants had experienced that they were not being treated with respect, and felt that people talked over their heads. These problems have been discussed in context of disability and aphasia, it affects social participation and with that, QoL in a negative way.[35 36] Raising awareness about LiS, both among caregivers and the public, is necessary, since it might lead to less insecurity and incertitude when meeting a person with LiS and, therefore result in people treating a person with LiS with more respect.

### Future research

From 2015, there will be an ICD-code in ICD-10-SE, which will improve the possibility of longitudinal research of this group of patients in Sweden. This seems to be an urgent field of research because, to our knowledge, there are no articles yet from Sweden about LiS. In addition, improving the development and research of new technical devices for this group of patients might help them to communicate more efficiently and to find methods of improving motor functions.

In the steering committee for the national quality register WebRehab, there are two representatives from patient organisations. There were no patient representatives involved in the recruitment to or conduction of the study. A short summary of the results has been distributed to the follow-up participants in the interviews.

**Acknowledgements** We would like to acknowledge the help from WebRehab and the clinics that provided information on the persons. At finally yet importantly, we would like to thank the participants for answering questionnaires. Sincere thanks to them and their families for allowing KS to visit them and for generously telling

about their situation and experiences. We would like to thank Kate Bramley-Moore for her English language assistance and proofreading.

**Contributors** KS, ÅN and KSS contributed to the design of the study. KS met the participants and gave them questionnaires, conducted and summarised the structured interviews. MT categorised the interviews inductively and wrote the manuscript together with KS and KSS.

**Funding** The study is in part funded by the Swedish Research Council (2017-00946) and a grant from the Swedish Heart and Lung Foundation (20160526).

**Disclaimer** The supporting bodies had no influence on the content of the study.

**Competing interests** None declared.

**Patient consent for publication** Not required.

**Ethics approval** The Regional Ethical Review Board in Gothenburg approved this study. The application was assigned number 052–15 and was accepted 20150415.

**Provenance and peer review** Not commissioned; externally peer reviewed.

**Data sharing statement** Due to ethical restrictions, data are available on request. Interested researchers may submit requests for data to the authors (contact KSS, email: ks.sunnerhagen@neuro.gu.se). According to Swedish regulation: http://www.epn.se/en/start/regulations/, the permission to use data is only for what has been applied for and then approved by the Ethical board. Not following the regulations is seen as scientific misconduct.

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
