## [Reviewer comments · BMJ Open]

ARTICLE DETAILS

TITLE (PROVISIONAL)	Locked-in syndrome in Sweden, an explorative study of persons who underwent rehabilitation: a cohort study
AUTHORS	Svernlings, Kajsa; Törnborn, Marie; Nordin, Åsa; Sunnerhagen, Katharina

VERSION 1 - REVIEW

REVIEWER	Antonio Cerasa IBFM-CNR, Catanzaro Italy
REVIEW RETURNED	11-Apr-2018

GENERAL COMMENTS	In this study Svernlings et al. are interested to describe the presence of the Locked-in syndrome in a Sweden population. Although this is a clinically relevant topic I have some difficulty in reading this paper in its present form 1) Authors speak about the role of rehabilitation for improving quality of life. I didn't find description of this rehabilitation. Which kind of rehabilitation has been performed?2) The narrative nature of the results section adds some "noise" while reading the text. One external reader would like to see a more scientific evaluation of clinical status.3) The qualitative investigation on only 10 patients can be considered a very explorative study.4) The authors did not provide any clinical and demographical information about Locked-in syndrome in order to allow comparison with other countries.5) The authors affirmed in pag. 9: "Statistical methods: for statistical analyses IBM® SPSS Statistics 21 was used. Mainly descriptive statistics with mean and median values were used." Please explain which statistical analysis has been performed. 1) Minor Points Please re-formulate the conclusion section in the Abstract. This is misleading. There are several typos throughout the manuscript Table 1: please remove "at 1.5.2015"
---

REVIEWER	Frank Becker University of Oslo Sunnaas Rehabilitation Hospital Norway I declare that I have collaborated with one of the authors (Katharina Stibrand Sunnerhagen) in a larger international project involving 9 centers in 7 countries (the Sunnaas International Network Stroke Study), resulting in 5 scientific papers in the years 2015 to 2018, in which we are co-authors.
REVIEW RETURNED	24-Oct-2018

GENERAL COMMENTS	The present study is a contribution to shed further light on a patient group where research still is sparse, locked-in syndrome. This is a rare condition, such that new investigations contribute to the field, even though they include lower numbers of patients as in this present study. In particular, long term follow up investigations – which is part of this paper – are warranted. This is especially true for environments and countries where no research on this patient group has been performed earlier, as the authors describe for Sweden. However, this kind of studies also contributes to the international literature by adding more cases to the, as mentioned, sparse literature (which I by the way think could be mentioned in the aims section which now is describing significance only for Sweden). My main objection to the present manuscript is that I miss a definition of locked-in syndrome (with regard to how the authors use this term in the manuscript) and a closer description of the patients' functional level. In order to compare and replicate the results, the paper should include more information about this, especially as there are different definitions and types of locked-in syndrome. A more detailed description can include information about functional level/type, time post stroke when diagnosed, inclusion criteria etc., as well as the "validation process" mentioned in the manuscript. Also, I suggest to consider whether it is possible to present the results in a more clearer way by reviewing which data are presented in which table and/or the text in the results section. In addition to this, I have a number of minor points and suggestions which are to be found below. Introduction I suggest a paragraph about different definitions of LIS. Material and Methods p6 line 47: "All were diagnosed with locked-in syndrome." Is there any more information on the diagnostic criteria? Does WebRehab have detailed criteria for locked-in syndrome? Which type of LIS? p 7 line 12: "data from medical charts were gathered for validation and quality control of a quality register." p 7 line 34: "For validation of the data in WebRehab" also legend figure: "excluded after the validation process due to not meeting the inclusion criteria (locked in syndrome)" (c.f. also p 20 line 19) – What did the validation process include? How was it performed? Why was it necessary? Does "quality register" refer to "WebRehab"? p 9 line 8 please explain the abbreviation "IPA-E" when used for the first time "is reliable for use in stroke patients" – but LIS-patients? Results
---

In general, please review how/where the data are presented; some data are presented in both text and 2 tables which seems unnecessary.

Table 1:

I suggest to add “only” to “* Deceased study persons included” and “** Alive study persons included”

“Classification of LIS at onset” – meaning at admission stroke unit or admission rehabilitation?

Length of stay – is this rehabilitation only? – is this total length of stay (stroke unit + rehab?)

p 11 line 23: “Time from onset to date of death varied from 1.5 to 2.3 years with a median of 1.9 years.” – this regards 3 persons: rather report when the 3rd person died, than the median (which actually is the same: 1,9 years)

p 11 line 25: “The cause of death was different for each case: pulmonary embolism, acute myocardial infarction, and acute vascular disorders of the intestine.” somewhat odd phrasing; although there were different causes, I think it is interesting that all were thrombo-embolic accidents, and that infection not caused death in this sample.

I find the distinction between “all study persons” and “participants” somewhat special and not the least difficult to follow throughout the manuscript which term refers to which group; maybe the latter could be called “who underwent follow-up”?

Table 2: In some cells, I cannot read all the words

Table 3:

What does “Alphabet board, blinking” mean? Does this refer to 2 methods, or one? Is the use of the alphabet board based on eye gaze in different directions?

What is the difference between “Alphabet board, blinking” and “Alphabet board (blinking)”?

Are these four patients in LIS now? Which type?

In first column, I suggest: “amount of assistance”, “communication type”

It would be interesting (at least for the 4 subjects who were followed up), to get information on how long each of them has lived with LIS.

Table 7:

Is there something wrong with the reference values both for Index value and VAS?

p 16 line 36: I do not quite understand the sentence “This information and support was also found by this participant by becoming a member of a neurological association.”

p 17 line 54 “Additional help from a nursing home was also reported.” I suspect that this refers to subject no. 2 – but this subject did get assistance from the nursing home only? I.e. not “additional” help?

p 18 line 7 “Every second week both an occupational therapist and a physiotherapist visited one participant, she still wanted a lot more additional training on top of these visits.” Was this the only subject receiving PT or OT? The phrasing suggests this to me.

Discussion

p 19 line 35: “We identified ten persons who had been diagnosed with LIS between 2007 and 2014” According to figure 1, 12 persons with LIS were identified?

p 20 line 32: “Another possible explanation for the small number of persons identified is that patients with LIS were not diagnosed and registered.” Is this different from p 20 line 19 “Another could be

	that incidence numbers are similar but all persons with LiS in Sweden have not been identified.”? And are these sentences in line with this statement under “strengths & limitatuons”: “study population constituted the whole population of patients with LiS”? p 22 line 3: “fact that LiS can be mistaken for other Disorders of Consciousness” and line 7 “or that the condition was misdiagnosed as another DOC” – please omit “other” and “another” here – as LIS is not a disorder of consciousness. p 23 line 5 “From 2015 there will be an ICD-code in ICD-10-SE,” => “Since ... there is”? Results on social participation (IPA, RAND 36) could possibly be shortly commented/discussed in the discussion section. language: Strengths & limitations : “worth noting is that most studies are fewer than ten or even single cases” => “have included fewer than 10 subjects or are single case studies” “and the results to other cultural contexts needs consideration.” lacks a word (“transferring”?) p 5 line 22: “they are both values that can differ, throughout life (19) which means it can be reduced in one” – please rephrase p 5 line 31 “Despite researchers belie” => “researchers” p 8 line 7: “and thereafter attempts to reach the study persons or proxy was by phone.” p 8 line 45: “for use on stroke patients” => “in” p 16 line 27: “participants described how there was a lack of information” => “participants described a lack of information” p 18 line 6 “They appreciated” => “The subjects”, “The interviewed persons with LIS” ... (se also p 18 line 40) p 19 line 8: “sometimes wives worked as personal assistants” => “in two of the cases, wives ...” (according to table 3) p 21 line 53: “and that fatality in LiS have declined” => “has” p 24 line 27: “A short summary of the results have been distributed to” => “has”
--	--

VERSION 1 – AUTHOR RESPONSE

Reviewer(s)' Comments to Author:

Reviewer Name: Antonio Cerasa

Institution and Country: IBFM-CNR, Catanzaro Italy

Please state any competing interests or state ‘None declared’: None to declare

Thank you for reviewing this paper. We have read your comments and tried to alter the paper accordingly. The answers are in *Italic* and the changes in the paper shown in highlighted yellow.

Reviewer: 1

Please leave your comments for the authors below

In this study Svernlín et al. are interested to describe the presence of the Locked-in syndrome in a Sweden population. Although this is a clinically relevant topic I have some difficulty in reading this paper in its present form

1) Authors speak about the role of rehabilitation for improving quality of life. I didn't find a description of this rehabilitation. Which kind of rehabilitation has been performed?

The patients had all been admitted to different specialized rehabilitation medicine centers in Sweden. Therefore we could identify them through the quality register. This information has been added in the text. In Sweden, the rehabilitation is individualized and goal-oriented. Exactly what each participant has received is not possible to tell. However, the focus has been on getting a functional communication, see to that the person gets nutrition in a safe manner, seating position in wheelchairs and of course physical therapy.

All the rehabilitation medicine clinics in Sweden participate in this quality register. (page 4-5)

All were diagnosed with locked-in syndrome and had been admitted for in-patient rehabilitation in specialized rehabilitation medicine clinics. (page 6-7)

2) The narrative nature of the results section adds some "noise" while reading the text. One external reader would like to see a more scientific evaluation of clinical status.

The narrative nature is in the part where the results of the qualitative interviews are presented. Since there are only 4 interviews and not very rich in content due to communicative difficulties, it is difficult to present it in another way.

3) The qualitative investigation on only 10 patients can be considered a very explorative study.

Yes, agree, and the qualitative interviews were only performed in 4 patients.

4) The authors did not provide any clinical and demographical information about Locked-in syndrome in order to allow comparison with other countries.

LiS is now described in the introduction. On p 5, we state that the situation of LiS in Sweden is unknown which therefore does not allow for comparison with other countries. We presented information on the patients in table 1.

5) The authors affirmed in pag. 9: "Statistical methods: for statistical analyses IBM® SPSS Statistics 21 was used. Mainly descriptive statistics with mean and median values were used." Please explain which statistical analysis has been performed.

There are no statistical analyses that have been performed except for calculating mean and median values for the description of demographics. The sample is too small to allow anything else.

1) Minor Points

Please re-formulate the conclusion section in the Abstract. This is misleading.

We have changed the last sentence to better reflect the results from this study.

In this study, the persons with LiS that were interviewed expressed the need for proper care, appropriate technical aids and a supportive environment in order to have quality of life. However, there is still much room for improvements. (p2)

There are several typos throughout the manuscript

We have gone through the manuscript and tried to address this.

Table 1: please remove “at 1.5.2015”

Done!

Reviewer: 2

Reviewer Name: Frank Becker

Institution and Country: University of Oslo, Sunnaas Rehabilitation Hospital, Norway

Please state any competing interests or state ‘None declared’: I declare that I have collaborated with one of the authors (Katharina Stibrand Sunnerhagen) in a larger international project involving 9 centers in 7 countries (the Sunnaas International Network Stroke Study), resulting in 5 scientific papers in the years 2015 to 2018, in which we are co-authors.

Thank you for reviewing this paper. The comments were valuable and we have tried to follow them. The answers to your questions/comments are in *Italic* and the changes in the paper shown in highlighted yellow.

Please leave your comments for the authors below

The present study is a contribution to shed further light on a patient group where research still is sparse, locked-in syndrome. This is a rare condition, such that new investigations contribute to the field, even though they include lower numbers of patients as in this present study. In particular, long term follow up investigations – which is part of this paper – are warranted. This is especially true for environments and countries where no research on this patient group has been performed earlier, as the authors describe for Sweden. However, this kind of studies also contributes to the international literature by adding more cases to the, as mentioned, sparse literature (which I by the way think could be mentioned in the aims section which now is describing significance only for Sweden).

As suggested, this has been added in the aims section. (p6)

Internationally the information on LiS is also sparse and in particular, the long term situation for them. This study will contribute with some more information regarding LiS.

My main objection to the present manuscript is that I miss a definition of locked-in syndrome (with regard to how the authors use this term in the manuscript) and a closer description of the patients’ functional level. In order to compare and replicate the results, the paper should include more information about this, especially as there are different definitions and types of locked-in syndrome. A more detailed description can include information about functional level/type, time post stroke when diagnosed, inclusion criteria etc., as well as the “validation process” mentioned in the manuscript.

Also, I suggest to consider whether it is possible to present the results in a more clearer way by reviewing which data are presented in which table and/or the text in the results section.

In addition to this, I have a number of minor points and suggestions which are to be found below.

Introduction

I suggest a paragraph about different definitions of LIS.

This has been added

The American Congress of Rehabilitation Medicine recommends neurobehavioral criteria to be used when diagnosing LiS. The criteria are 1: Eye opening well sustained, 2: Basic cognitive abilities

preserved (clinical examination), 3: Severe hypophonia or aphonia on clinical examination, 4: Quadriplegia/Quadriparesis on clinical examination and 5: Communication primarily through eye movements or through blinking. The LiS can vary in clinical presentation. Classic LiS is defined as a fully paralyzed patient with intact vertical eye movements and movement in the eyelid. Incomplete LiS is similar to Classic LiS but with remnants of motor functions beyond those of the classic variant, usually some movement in fingers and/or toes. Total LiS is defined as total immobility, the use of electroencephalography – EEG is then necessary to ascertain consciousness. In context of duration, LiS can be chronic or transient, in the latter the patient recovers completely.

Material and Methods

p6 line 47: “All were diagnosed with locked-in syndrome.” Is there any more information on the diagnostic criteria? Does WebRehab have detailed criteria for locked-in syndrome? Which type of LIS?

The definition used is the following:

Locked-in syndrome

A condition caused by interrupted corticospinal and corticobulbar connections.

The patient has tetraplegia and cannot use speech to communicate but is fully preserved cognitively, there is retained eye opening please exclude bilateral ptos that may complicate the condition), afoni or hypophony tetraplegia or tetrapares

Can adequately communicate yes / no using vertical or horizontal eye movements alternatively eye blinking

p 7 line 12: “data from medical charts were gathered for validation and quality control of a quality register.” p 7 line 34: “For validation of the data in WebRehab”

also legend figure: “excluded after the validation process due to not meeting the inclusion criteria (locked in syndrome)” (c.f. also p 20 line 19)

– What did the validation process include? How was it performed? Why was it necessary? Does “quality register” refer to “WebRehab”?

To make certain that the data in WebRehab were entered correctly (a validation), the medical charts were checked.

p 9 line 8 please explain the abbreviation “IPA-E” when used for the first time

“is reliable for use in stroke patients” – but LIS-patients?

IPA-E is explained on p 8. The IPA-E is not said to be reliable for use in stroke patients. It is developed to be used to assess the situation of long-term conditions living in the community (not hospitalized). The SIS however, is reliable for stroke patients but it was not clear before, that it was developed for stroke.

The SIS is developed for, validated and shown to be reliable for stroke patients

Results

In general, please review how/where the data are presented; some data are presented in both text and 2 tables which seems unnecessary.

Thank you for noticing that. We have tried to delete the information in the text that were repetitive of the tables.

Table 1:

I suggest to add “only” to “* Deceased study persons included” and “** Alive study persons included”

Done as suggested.

“Classification of LiS at onset” – meaning at admission stroke unit or admission rehabilitation?

This has been changed to make it clearer.

Classification of LiS at admittance for rehabilitation

Length of stay – is this rehabilitation only? – is this total length of stay (stroke unit + rehab?)

Thank’s for asking this. We have now altered to include total stay (intensive care+stroke unit+rehab)

To make certain that the data in WebRehab had been entered correctly (a validation), the medical charts were checked.

p 11 line 23: “Time from onset to date of death varied from 1.5 to 2.3 years with a median of 1.9 years.” – this regards 3 persons: rather report when the 3rd person died, than the median (which actually is the same: 1,9 years)

Yes this was not well written and have been changed.

One person died during rehabilitation and the remaining two after the initial rehabilitation period, time from onset to date of death was 1.5, 1.9 and 2.3 years.

p 11 line 25: “The cause of death was different for each case: pulmonary embolism, acute myocardial infarction, and acute vascular disorders of the intestine.” somewhat odd phrasing; although there were different causes, I think it is interesting that all were thrombo-embolic accidents, and that infection not caused death in this sample.

Agree- we have rephrased this to

The cause of death was pulmonary embolism,

I find the distinction between “all study persons” and “participants” somewhat special and not the least difficult to follow throughout the manuscript which term refers to which group; maybe the latter could be called “who underwent follow-up”?

We have changed the wording in most places to follow-up participants when appropriate (in the text as well as the heading in table 3)

Table 2: In some cells, I cannot read all the words

Thanks for noticing that- the cells had shifted

Table 3:

What does “Alphabet board, blinking” mean? Does this refer to 2 methods, or one? Is the use of the alphabet board based on eye gaze in different directions?

Aphabet board or blinking equal time. Yes, it is based on eye gaze in different directions

What is the difference between “Alphabet board, blinking” and “Alphabet board (blinking)”?

The second is Alphabet board mainly

Are these four patients in LIS now? Which type?

At the time of examination they all were able to communicate to some extent which actually means that they no longer are locked in. However, they defined themselves as having locked-in syndrome.

In first column, I suggest: “amount of assistance”, “communication type”

It would be interesting (at least for the 4 subjects who were followed up), to get information on how long each of them has lived with LIS.

This has been added in table 3 (ranged from 5.5 years to 8)

Table 7:

Is there something wrong with the reference values both for Index value and VAS?

Yes, there was a mistake

VAS should read 62.7 (58.8-66.6)

p 16 line 36: I do not quite understand the sentence “This information and support was also found by this participant by becoming a member of a neurological association.”

Has been changed (it is called Neuro association)

member of a patient association.

p 17 line 54 “Additional help from a nursing home was also reported.” I suspect that this refers to subject no. 2 – but this subject did get assistance from the nursing home only? I.e. not “additional” help?

This was deleted- this was another person who lived close to a nursing home

p 18 line 7 “Every second week both an occupational therapist and a physiotherapist visited one participant, she still wanted a lot more additional training on top of these visits.” Was this the only subject receiving PT or OT? The phrasing suggests this to me.

Yes.

Discussion

p 19 line 35: “We identified ten persons who had been diagnosed with LiS between 2007 and 2014” According to figure 1, 12 persons with LIS were identified?

12 were entered as LiS in WebRehab, one was not correct after having read the medical charts and one could not be identified due to administrative mistake.

One person was excluded after the validation process due to not meeting the inclusion criteria and one person due to not having a valid personal identity number.

p 20 line 32: “Another possible explanation for the small number of persons identified is that patients with LiS were not diagnosed and registered.” Is this different from p 20 line 19 “Another could be that incidence numbers are similar but all persons with LiS in Sweden have not been identified.”?

We have changed this and deleted the last sentence

And are these sentences in line with this statement under “strengths & limitations”: “study population constituted the whole population of patients with LIS”?

You are of course right that this does not make sense. We have altered the sentence as follows

The study population constituted the whole population of patients with LIS receiving in-patient rehabilitation between 2007-2014.

p 22 line 3: “fact that LIS can be mistaken for other Disorders of Consciousness” and line 7 “or that the condition was misdiagnosed as another DOC” – please omit “other” and “another” here – as LIS is not a disorder of consciousness.

Agree and this has been deleted.

p 23 line 5 “From 2015 there will be an ICD-code in ICD-10-SE,” => “Since ... there is”?

Results on social participation (IPA, RAND 36) could possibly be shortly commented/discussed in the discussion section.

language:

Strengths & limitations :

“worth noting is that most studies are fewer than ten or even single cases” => “have included fewer than 10 subjects or are single case studies”

“and the results to other cultural contexts needs consideration.” lacks a word (“transferring”?)

p 5 line 22: “they are both values that can differ, throughout life (19) which means it can be reduced in one” – please rephrase

they are values that can differ, throughout life

p 5 line 31 “Despite researchers belief” => “researchers”

Done

p 8 line 7: “and thereafter attempts to reach the study persons or proxy was by phone.”

thereafter the attempts to reach the study persons or proxy was by phone

p 8 line 45: “for use on stroke patients” => “in”

not there any more

p 16 line 27: “participants described how there was a lack of information” => “participants described a lack of information”

done

p 18 line 6 “They appreciated” => “The subjects”, “The interviewed persons with LIS” ... (se also p 18 line 40)

Changed

p 19 line 8: "sometimes wives worked as personal assistants" => "in two of the cases, wives ..."
(according to table 3)

done

p 21 line 53: "and that fatality in LiS have declined" => "has"

done

p 24 line 27: "A short summary of the results have been distributed to" => "has"

done

VERSION 2 – REVIEW

REVIEWER	Frank Becker Sunnaas Rehabilitation Hospital, Nesoddtangen, Norway University of Oslo, Oslo, Norway I declare that I have collaborated with one of the authors (Katharina Stibrand Sunnerhagen) in a larger international project involving 9 centers in 7 countries (the Sunnaas International Network Stroke Study), resulting in 5 scientific papers in the years 2015 to 2018, in which we are co-authors.
REVIEW RETURNED	13-Jan-2019

GENERAL COMMENTS	I thank the authors for the revised manuscript and their responses to my comments and concerns. The manuscript has been substantially revised and improved, and I have now only some minor comments which I recommend the authors to take a look at. Definition of locked-in syndrome (page 4 of the revised manuscript): Here it is stated that "In context of duration, LiS can be chronic or transient, in the latter the patient recovers completely". In my understanding of the term "transient LIS", it does not only describe complete, but also partial recovery. Thus, it is used when patients experience recovery so that they not anymore fulfil the criteria for LIS, but still might have (often substantial) sequelae due to their stroke. Please explain or rephrase. As requested in my review, the authors have added a paragraph on definition and classification of LIS – thank you for that. I support the choice of the ACRM criteria. However, for the inclusion to this study, the criteria for LIS from the WebRehab registry were used (page 7, line 44ff). These are explained in the response to review, but not in the manuscript. I would find it helpful if these criteria were mentioned either in the introduction or the methods section. Table 1: Length of stay: If I understand correctly, the median stay in the acute phase (i.e. stroke unit) was 163 – 151 = 12 days. This seems rather short, given the fact that LIS-patients often have complications (e.g. regarding respiration) in the early phase? The upper range for length of total stay is stated to be 449 days. On page 12 line 29 it says "One person died during rehabilitation ... , time from onset to date of death was 1.5, 1.9 and 2.3 years."
--

	Should this not mean that the upper range of total length of stay at least was 1.5 years = 548 days? Or maybe I misunderstand something here, maybe patients had periods with neither acute care nor rehab? It might possibly be helpful to state very short in the methods section how length of stay, in rehab and totally, was calculated. Table 2: Footnote 1 (“Full years”) – is this footnote actually used in the table? Table 7: For Index value, it says (mean, 95 % CI): 0.44 (0.28-0.42). This does not seem correct? Page 17, lines 37 ff: “One person had some oral communication including most vowels, some consonants and a few short words, today he can control his jaw and tongue muscles and he has some movement in his fingers.” I understand this to mean that only one person had some oral communication, while it is stated in the Response to review: “At the time of examination they all were able to communicate to some extent” – how does this fit together? page 18, lines 7 ff “The interviewed persons with LIS appreciated having phenomenal physiotherapists”. The next sentence is “Every second week both an occupational therapist and a physiotherapist visited one participant”. Upon a question of mine, the authors confirm in their Response to review that only 1 of the 4 interviewed persons receives PT or OT. This does not seem to be in agreement with the plural “phenomenal physiotherapists”? Also, I have some minor suggestions regarding language; they are tracked in the attached Word-file. The reviewer provided a marked copy with additional comments. Please contact the publisher for full details.
--	--

VERSION 2 – AUTHOR RESPONSE

Reviewer(s)' Comments to Author:

Reviewer: 2

Reviewer Name: Frank Becker

Institution and Country: Sunnaas Rehabilitation Hospital, Nesoddtangen, Norway University of Oslo, Oslo, Norway Please state any competing interests or state ‘None declared’:

I declare that I have collaborated with one of the authors (Katharina Stibrand Sunnerhagen) in a larger international project involving 9 centers in 7 countries (the Sunnaas International Network Stroke Study), resulting in 5 scientific papers in the years 2015 to 2018, in which we are co-authors.

Please leave your comments for the authors below I thank the authors for the revised manuscript and their responses to my comments and concerns. The manuscript has been substantially revised and

improved, and I have now only some minor comments which I recommend the authors to take a look at.

Definition of locked-in syndrome (page 4 of the revised manuscript):

Here it is stated that “In context of duration, LiS can be chronic or transient, in the latter the patient recovers completely”. In my understanding of the term “transient LIS”, it does not only describe complete, but also partial recovery. Thus, it is used when patients experience recovery so that they not anymore fulfil the criteria for LIS, but still might have (often substantial) sequelae due to their stroke. Please explain or rephrase.

This has been rephrased.

In the latter, the patients recover so that they not anymore fulfil the criteria for LIS, but still might have (often substantial) sequelae due to their stroke.

As requested in my review, the authors have added a paragraph on definition and classification of LIS – thank you for that. I support the choice of the ACRM criteria. However, for the inclusion to this study, the criteria for LIS from the WebRehab registry were used (page 7, line 44ff). These are explained in the response to review, but not in the manuscript. I would find it helpful if these criteria were mentioned either in the introduction or the methods section.

The text in WebRehab that helps the physician to identify a LiS patient has been added in the methods part, under study population.

Table 1:

Length of stay: If I understand correctly, the median stay in the acute phase (i.e. stroke unit) was 163 – 151 = 12 days. This seems rather short, given the fact that LIS-patients often have complications (e.g. regarding respiration) in the early phase?

No this is a misunderstanding. 1 patient had that short period of stay in the stroke unit. The median for the whole hospital stay is given as well as the whole stay for rehab.

The upper range for length of total stay is stated to be 449 days. On page 12 line 29 it says “One person died during rehabilitation ... , time from onset to date of death was 1.5, 1.9 and 2.3 years.” Should this not mean that the upper range of total length of stay at least was 1.5 years = 548 days? Or maybe I misunderstand something here, maybe patients had periods with neither acute care nor rehab?

This is a mistake from our side when entering the dates in the file

It might possibly be helpful to state very short in the methods section how length of stay, in rehab and totally, was calculated.

This is not done. It is calculated based on dates.

Table 2:

Footnote 1 (“Full years”) – is this footnote actually used in the table?

Thank you for noticing this. This is left from an earlier version of the table!

Table 7:

For Index value, it says (mean, 95 % CI): 0.44 (0.28-0.42). This does not seem correct?

You are absolutely right (the CI was for women and not for the whole population. This is now corrected in table 7

0.44 (0.38-0.50)

Page 17, lines 37 ff: "One person had some oral communication including most vowels, some consonants and a few short words, today he can control his jaw and tongue muscles and he has some movement in his fingers." I understand this to mean that only one person had some oral communication, while it is stated in the Response to review: "At the time of examination they all were able to communicate to some extent" – how does this fit together?

This has been re-written. The sentence was to show improvement in communication since discharge.

One person had some oral communication including most vowels, some consonants and a few short words at discharge. At the time of assessment, he could control his jaw and tongue muscles and he had some movement in his fingers.

page 18, lines 7 ff "The interviewed persons with LIS appreciated having phenomenal physiotherapists". The next sentence is "Every second week both an occupational therapist and a physiotherapist visited one participant". Upon a question of mine, the authors confirm in their Response to review that only 1 of the 4 interviewed persons receives PT or OT.

This does not seem to be in agreement with the plural "phenomenal physiotherapists"?

Again, we were not clear.

phenomenal physiotherapists during the rehabilitation period.